# Learning Actionable Representations with Goal-Conditioned Policies

**Dibya Ghosh, Abhishek Gupta, & Sergey Levine** [*]
Department of Electrical Engineering and Computer Science
University of California, Berkeley
Berkeley, CA 94703, USA

## Abstract

Representation learning is a central challenge across a range of machine learning areas. In reinforcement learning, effective and functional representations have the potential to tremendously accelerate learning progress and solve more challenging problems. Most prior work on representation learning has focused on generative approaches, learning representations that capture all the underlying factors of variation in the observation space in a more disentangled or well-ordered manner. In this paper, we instead aim to learn functionally salient representations: representations that are not necessarily complete in terms of capturing all factors of variation in the observation space, but rather aim to capture those factors of variation that are important for decision making – that are "actionable." These representations are aware of the dynamics of the environment, and capture only the elements of the observation that are necessary for decision making rather than all factors of variation, eliminating the need for explicit reconstruction. We show how these learned representations can be useful to improve exploration for sparse reward problems, to enable long horizon hierarchical reinforcement learning, and as a state representation for learning policies for downstream tasks. We evaluate our method on a number of simulated environments, and compare it to prior methods for representation learning, exploration, and hierarchical reinforcement learning.

## 1 Introduction

Representation learning refers to a transformation of an observation, such as a camera image or state observation, into a form that is easier to manipulate to deduce a desired output or perform a downstream task, such as prediction or control. In reinforcement learning (RL) in particular, effective representations are ones that enable generalizable controllers to be learned quickly for challenging and temporally extended tasks. While end-to-end representation learning with full supervision has proven effective in many scenarios, from supervised image recognition (Krizhevsky et al., 2012) to vision-based robotic control (Levine et al., 2015), devising representation learning methods that can use unlabeled data or experience effectively remains an open problem.

Much of the prior work on representation learning in RL has focused on generative approaches. Learning these models is often challenging because of the need to model the interactions of *all* elements of the state. We instead aim to learn functionally salient representations: representations that are not necessarily complete in capturing all factors of variation in the observation space, but rather aim to capture factors of variation that are relevant for decision making – that are actionable.

How can we learn a representation that is aware of the dynamical structure of the environment? We propose that a basic understanding of the world can be obtained from a *goal-conditioned policy*, a policy that can knows how to reach arbitrary goal states from a given state. Learning how to execute shortest paths between all pairs of states suggests a deep understanding of the environment dynamics, and we hypothesize that a representation incorporating the knowledge of a goal-conditioned policy can be readily used to accomplish more complex tasks. However, such a policy does not provide a readily usable state representation, and it remains to choose *how* an effective state representation

---

[*] dibya.ghosh@berkeley.edu

should be extracted. We want to extract those factors of the state observation that are critical for deciding which action to take. We can do this by comparing which actions a goal-conditioned policy takes for two different goal states. Intuitively, if two goal states require different actions, then they are functionally different and vice-versa. This principle is illustrated in the diagram in Figure 1. Based on this principle, we propose actionable representations for control (ARC), representations in which Euclidean distances between states correspond to expected differences between actions taken to reach them. Such representations emphasize factors in the state that induce significant differences in the corresponding actions, and de-emphasize those features that are irrelevant for control.

While learning a goal-conditioned policy to extract such a representation might itself represent a daunting task, it is worth noting that such a policy can be learned without any knowledge of downstream tasks, simply through unsupervised exploration of the environment. It is reasonable to postulate that, without active exploration, *no* representation learning method can possibly acquire a dynamics-aware representation, since understanding the dynamics requires experiencing transitions and interactions, rather than just observations of valid states. As we demonstrate in our experiments, representations extracted from goal-conditioned policies can be used to better learn more challenging tasks than simple goal reaching, which cannot be easily contextualized by goal states. The process of learning goal-conditioned policies can also be made recursive, so that the actionable representations learned from one goal-conditioned policy can be used to quickly learn a better one.

Actionable representations for control are useful for a number of downstream tasks: as representations for task-specific policies, as representations for hierarchical RL, and to construct well-shaped reward functions. We show that ARCs enable these applications better than representations that are learned using unsupervised generative models, predictive models, and other prior representation learning methods. We analyze structure of the learned representation, and compare the performance of ARC with a number of prior methods on downstream tasks in simulated robotic domains such as wheeled locomotion, legged locomotion, and robotic manipulation.

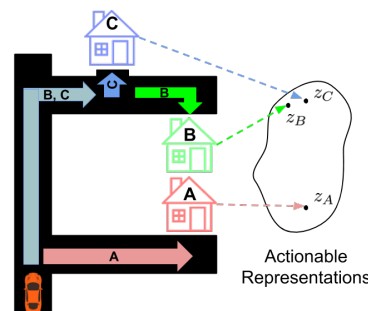

Figure 1: Actionable representations: 3 houses A, B, C can only be reached by indicated roads. The actions taken to reach A, B, C are shown by arrows. Although A, B are very close in space, they are functionally different. The car has to take a completely different road to reach A, compared to B and C. Representations $z_A$, $z_B$, $z_C$ learn these functional differences to differentiate A from B and C, while keeping B and C close.

## 2 PRELIMINARIES

**Goal-conditioned reinforcement learning.** In RL, the goal is to learn a policy $\pi_\theta(a_t|s_t)$ that maximizes the expected return $R_t = \mathbb{E}_{\pi_\theta}[\sum_t r_t]$. Typically, RL learns a single task that optimizes for a particular reward function. If we instead would like to train a policy that can accomplish a variety of tasks, we might instead train a policy that is conditioned on another input – a goal. When the different tasks directly correspond to different states, this amounts to conditioning the policy $\pi$ on both the current and goal state. The policy $\pi_\theta(a_t|s_t, g)$ is trained to reach goals from the state space $g \sim \mathcal{S}$, by optimizing $\mathbb{E}_{g \sim S}[\mathbb{E}_{\pi_\theta(a|s,g)}(R_g))]$, where $R_g$ is a reward for reaching the goal $g$.

**Maximum entropy RL.** Maximum entropy RL algorithms modify the RL objective, and instead learns a policy to maximize the reward as well as the entropy of the policy (Haarnoja et al., 2017; Todorov, 2006), according to $\pi^\star = \arg\max_\pi E_\pi[r(s, a)] + \mathcal{H}(\pi)$. In contrast to standard RL, where optimal policies in fully observed environments are deterministic, the solution in maximum entropy RL is a stochastic policy, where the entropy reflects the sensitivity of the rewards to the action: when the choice of action has minimal effect on future rewards, actions are more random, and when the choice of action is critical, the actions are more deterministic. In this way, the action distributions for a maximum entropy policy carry more information about the dynamics of the task.

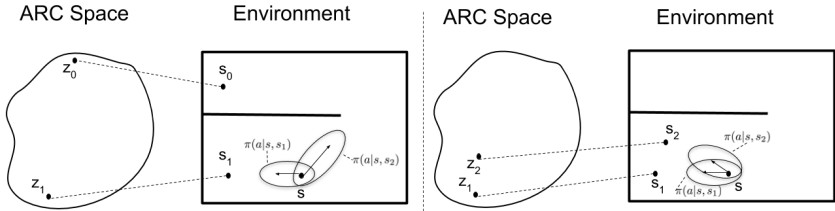

Figure 2: An illustration of actionable representations. For a pair of states $s_1, s_2$, the divergence between the goal-conditioned action distributions they induce defines the actionable distance $D_{\text{Act}}$, which in turn is used to learn representation $\phi$.

## 3 LEARNING ACTIONABLE REPRESENTATIONS

In this work, we extract a representation that can distinguish states based on actions required to reach them, which we term an actionable representation for control (ARC). In order to learn state representations $\phi$ that can capture the elements of the state which are important for decision making, we first consider defining actionable distances $D_{\text{Act}}(s_1, s_2)$ between states. Actionable distances are distances between states that capture the differences between the actions required to reach the different states, thereby implicitly capturing dynamics. If actions required for reaching state $s_1$ are very different from the actions needed for reaching state $s_2$, then these states are *functionally* different, and should have large actionable distances. This subsequently allows us to extract a feature representation($\phi(s)$) of state, which captures elements that are important for decision making.

To formally define actionable distances, we build on the framework of goal-conditioned RL. We assume that we have already trained a maximum entropy goal-conditioned policy $\pi_\theta(a|s, g)$ that can start at an arbitrary state $s_0 \in S$ in the environment, and reach a goal state $s_g \in \mathcal{S}$. Although this is a significant assumption, we will discuss later how this is in fact reasonable in many settings. We can extract actionable distances by examining how varying the goal state affects action distributions for goal-conditioned policies. Formally, consider two different goal states $s_1$ and $s_2$. At an intermediate state $s$, the goal-conditioned policy induces different action distributions $\pi_\theta(a|s, s_1)$ and $\pi_\theta(a|s, s_2)$ to reach $s_1$ and $s_2$ respectively. If these distributions are similar over many intermediate states $s$, this suggests that these states are functionally similar, while if these distributions are different, then the states must be functionally different. This motivates a definition for actionable distances $D_{\text{Act}}$ as

$$D_{\text{Act}}(s_1, s_2) = \mathbb{E}_s \left[ D_{KL}(\pi(a|s, s_1)||\pi(a|s, s_2)) + D_{KL}(\pi(a|s, s_2)||\pi(a|s, s_1)) \right]. \quad (1)$$

The distance consists of the expected divergence over *all* initial states $s$ (refer to Section B for how we do this practically). If we focus on a subset of states, the distance may not capture action differences induced elsewhere, and can miss functional differences between states. Since maximum entropy policies learn *unique* optimal stochastic policies, the actionable distance is well-defined and unambiguous. Furthermore, because max-ent policies capture sensitivity of the value function, goals are similar under ARC if they require the same action *and* they are equally "easy" to reach.

We can use $D_{\text{Act}}$ to extract an actionable representation of state. To learn this representation $\phi(s)$, we optimize $\phi$ such that Euclidean distance between states in representation space corresponds to actionable distances $D_{\text{Act}}$ between them. This optimization yields good representations of state because it emphasizes the functionally relevant elements of state, which significantly affect the actionable distance, while suppressing less functionally relevant elements of state. The problem is:

$$\min_\phi \mathbb{E}_{s_1, s_2} \left[ \|\phi(s_1) - \phi(s_2)\|_2 - D_{\text{Act}}(s_1, s_2) \right]^2 \quad (2)$$

This objective yields representations where Euclidean distances are meaningful. This is not necessarily true in the state space or in generative representations (Section 6.4). These representations are meaningful for several reasons. First, since we are leveraging a goal-conditioned policy, they are aware of dynamics and are able to capture local connectivity of the environment. Secondly, the representation is optimized so that it captures only the *functionally* relevant elements of state.

**Requirement for Goal Conditioned Policy:**    A natural question to ask is whether needing a goal-conditioned policy is too strong of a prerequisite. However, it is worth noting that the GCP can be trained with existing RL methods (TRPO) using a sparse task-agnostic reward (Section 6.2, Appendix A.1) – obtaining such a policy is not especially difficult, and existing methods are quite capable of doing so ( Nair et al. (2018)). Furthermore, it is likely not possible to acquire a functionality-aware state representation *without* some sort of active environment interaction, since dynamics can only be understood by observing outcomes of actions, rather than individual states. Importantly, we discuss in the following section how ARCs help us solve tasks beyond what a simple goal-conditioned policy can achieve.

# 4    USING ACTIONABLE REPRESENTATIONS FOR DOWNSTREAM TASKS

A natural question that emerges when learning representations from a goal-conditioned policy pertains to what such a representation enables over the goal-conditioned policy itself. Although goal-conditioned policies enable reaching between arbitrary states, they suffer from fundamental limitations: they do not generalize very well to new states, and they are limited to solving only goal-reaching tasks. We show in our empirical evaluation that the ARC representation expands meaningfully over these limitations of a goal-conditioned policy - to new tasks and to new regions of the environment. In this section, we detail how ARCs can be used to generalize beyond a goal-conditioned policy to help solve tasks that cannot be expressed as goal reaching (Section 4.1), tasks involving larger regions of state space (Section 4.2), and temporally extended tasks which involve sequences of goals (Section 4.3).

## 4.1    FEATURES FOR LEARNING POLICIES

Goal-conditioned policies are trained with only a goal-reaching reward, and so are unaware of reward structures used for other tasks in the environment which do not involve simple goal-reaching. Tasks which cannot be expressed as simply reaching a goal are abundant in real life scenarios such as navigation under non-uniform preferences or manipulation with costs on quality of motion, and for such tasks, using the ARC representation as input for a policy or value function can make the learning problem easier. We can learn a policy for a downstream task, of the form $\pi_\theta(a|\phi(s))$, using the representation $\phi(s)$ instead of state $s$. The implicit understanding of the environment dynamics in the learned representation prioritizes the parts of the state that are most important for learning, and enables quicker learning for these tasks as we see in Section 6.6.

## 4.2    REWARD SHAPING

We can use ARC to construct better-shaped reward functions. It is common in continuous control to define rewards in terms of some distance to a desired state, oftentimes using Euclidean distance (Schulman et al., 2015; Lillicrap et al., 2015). However, Euclidean distance in state space is not necessarily a meaningful metric of *functional* proximity. ARC provides a better metric, since it directly accounts for reachability. We can use the actionable representation to define better-shaped reward functions for downstream tasks. We define a shaping of this form to be the negative Euclidean distance between two states in ARC space: $-||\phi(s_1) - \phi(s_2)||_2$: for example, on a goal-reaching task $r(s) = r_{\text{sparse}}(s, s_g) - ||\phi(s) - \phi(s_g)||_2$. This allows us to explore and learn policies even in the presence of sparse reward functions.

One may wonder whether, instead of using ARCs for reward shaping, we might directly use the goal-conditioned policy to reach a particular goal. As we will illustrate in Section 6.5, the representation typically generalizes better than the goal-conditioned policy. Goal-conditioned policies typically can be trained on small regions of the state space, but don't extrapolate well to new parts of the state space. We observe that ARC exhibits better generalization, and can provide effective reward shaping for goals that are very difficult to reach with the goal-conditioned policy.

## 4.3    HIERARCHICAL REINFORCEMENT LEARNING

Goal-conditioned policies can serve as low-level controllers for hierarchical tasks which require synthesizing a particular sequence of behaviours, and thus not expressible as a single goal-reaching

objective. One approach to solving such tasks learns a high-level controller $\pi_{\text{meta}}(g|s)$ via RL that produces desired goal states for a goal-conditioned policy to reach sequentially (Nachum et al., 2018). The high-level controller suggests a goal, which the goal conditioned policy attempts to reach for several time-steps, following which the high-level controller picks a new goal. For many tasks, naively training such a high-level controller which outputs goals directly in state space is unlikely to perform well, since such a controller must disentangle the relevant attributes in the *goal* for long horizon reasoning. We consider two schemes to use ARCs for hierarchical RL - learning a high level policy which commands directly in ARC space or commands in a clustered latent space.

**HRL directly in ARC space:** ARC representations provide a better goal space for high-level controllers, since they de-emphasize components of the goal space irrelevant for determining the optimal action. In this scheme, the high-level controller $\pi_{\text{meta}}(z|s)$ observes the current state and generates a distribution over points in the latent space. At every meta-step, a sample $z_h$ is taken from $\pi_{\text{meta}}(z|s)$ which represents the high level command. $z_h$ is then translated into a goal $g_h$ via a decoder which is trained to reconstruct states from their corresponding goals. This goal $g_h$ can then be used to command the goal conditioned policy for several time steps, before resampling again from $\pi_{\text{meta}}$. A high-level controller producing outputs in ARC space does not need to rediscover saliency in the goal space, which makes the search problem less noisy and more accurate. We show in Section 4.3 that using ARC as a hierarchical goal space enables significant improvement for waypoint navigation tasks.

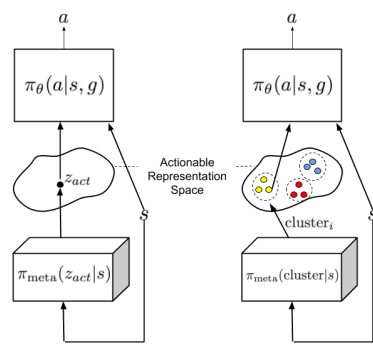

Figure 3: Hierarchical RL with ARC. **Left:** Directly commanding in ARC space **Right:** Commanding a cluster in ARC space

**Clustering in ARC space:** Since ARC captures the topology of the environment, clusters in ARC space often correspond to semantically meaningful state abstractions. We utilize these clusters, with the intuition that a meta-controller searching in "cluster space" should learn faster than directly outputting states. In this scheme, we first build a discrete number of clusters by clustering the points that the goal conditioned policy is trained on using the k-means algorithm within the ARC representation space. We then train a high-level controller $\pi_{\text{meta}}(c|s)$ which observes a state $s$ and generates a distribution over discrete clusters $c$. At every meta-step, a cluster sample $c_h$ is taken from $\pi_{\text{meta}}(c|s)$. A goal in state space $g_h$ is then chosen uniformly at random from points within the cluster $c_h$ and used to command the GCP for several time steps before the next cluster is sampled from $\pi_{\text{meta}}$. We train a meta-policy to output clusters, instead of states: $\pi_{meta}(\text{cluster}|s)$. We see that for hierarchical tasks with less granular reward functions such as room navigation, performing RL in "cluster space" induced by ARC outperform cluster spaces induced by other representations, since the distance metric is much more meaningful in ARC space.

## 5 RELATED WORK

The capability to learn effective representations is a major advantage of deep neural network models. These representations can be acquired implicitly, through end-to-end training (Goodfellow et al., 2016), or explicitly, by formulating and optimizing a representation learning objective. A classic approach to representation learning is generative modeling, where a latent variable model is trained to model the data distribution, and the latent variables are then used as a representation (Rasmus et al., 2015; Dumoulin et al., 2016; Kingma & Welling, 2013; Finn et al., 2015; Ghadirzadeh et al., 2017; Curran et al., 2015; Goroshin et al., 2015; Higgins et al., 2017). In the context of control and sequence models, generative models have also been proposed to model transitions (Watter et al., 2015; Assael et al., 2015; Zhang et al., 2018b; Kurutach et al., 2018). While generative models are general and principled, they must not only explain the entirety of the input observation, but must also generate it. Several methods perform representation learning without generation, often based on contrastive losses (Sermanet et al., 2018; van den Oord et al., 2018; Belghazi et al., 2018; Chopra et al., 2005; Weinberger & Saul, 2009). While these methods avoid generation, they either still require modeling of the entire input, or utilize heuristics that encode user-defined information.

In contrast, ARCs are directly trained to focus on decision-relevant features of input, providing a broadly applicable objective that is still selective about which aspects of input to represent.

In the context of RL and control, representation learning methods have been used for many downstream applications (Lesort et al., 2018), including representing value functions (Barreto et al., 2016) and building models (Watter et al., 2015; Assael et al., 2015; Zhang et al., 2018b). Our approach is complementary: it can also be applied to these applications. Several works have sought to learn representations that are specifically suited for physical dynamical systems (Jonschkowski & Brock, 2015) and that use interaction to build up dynamics-aware features (Bengio et al., 2017; Laversanne-Finot et al., 2018). In contrast to Jonschkowski & Brock (2015), our method does not attempt to encode all physically-relevant features of state, only those relevant for choosing actions. In contrast to Bengio et al. (2017); Laversanne-Finot et al. (2018), our approach does not try to determine which features of the state can be independently controlled, but rather which features are relevant for *choosing* controls. Srinivas et al. (2018) also consider learning representations through goal-directed behaviour, but receives supervision through demonstrations instead of active observation. Related methods learn features that are predictive of actions based on pairs of sequential states (so-called *inverse models*) (Agrawal et al., 2016; Pathak et al., 2017; Zhang et al., 2018a). More recent work such as (Burda et al., 2018b) perform a large scale study of these types of methods in the context of exploration. Unlike ARC, which is learned from a policy performing long-horizon control, inverse models are not obliged to represent all relevant features for multi-step control, and suffer from greedy reasoning.

## 6 EXPERIMENTS

The aim of our experimental evaluation is to study the following research questions:

1. Can we learn ARCs for multiple continuous control environments? What are the properties of these learned representations?
2. Can ARCs be used as feature representations for learning policies quickly on new tasks?
3. Can reward shaping with ARCs enable faster learning?
4. Do ARCs provide an effective mechanism for hierarchical RL?

Full experimental and hyperparemeter tuning details are presented in the appendix.

### 6.1 DOMAINS

We study six simulated environments as illustrated in Figure 4: 2D navigation tasks in two settings, wheeled locomotion tasks in two settings, legged locomotion, and object pushing with a robotic gripper. The 2D navigation domains consist of either a room with a central divider wall or four rooms. Wheeled locomotion involves a two-wheeled differential drive robot, either in free space or with four rooms. For legged locomotion, we use a quadrupedal ant robot, where the state space consists of all joint angles, along with the Cartesian position of the center of mass (CoM). The manipulation task uses a simulated Sawyer arm to push an object, where the state consists of end-effector and object positions. Further details are presented in Appendix C.

These environments present interesting representation learning challenges. In 2D navigation, the walls impose structure similar to those in Figure 1: geometrically proximate locations on either side of a wall are far apart in terms of reachability. The locomotion environments present an additional challenge: an effective representation must account for the fact that the internal joints of each robot (legs or wheel orientation) are less salient for long-horizon tasks than CoM. The original state representation does not reflect this structure: joint angles expressed in radians carry as much weight as CoM positions in meters. In the object manipulation task, a key representational challenge is to distinguish between pushing the block and simply moving the arm in free space.

### 6.2 LEARNING THE GOAL-CONDITIONED POLICY AND ARC REPRESENTATION

We first learn a stochastic goal-conditioned policy parametrized by a neural network which outputs actions given the current state and the desired goal. This goal-conditioned policy is trained using a sparse reward using entropy-regularized Trust Region Policy Optimization (TRPO) (Schulman

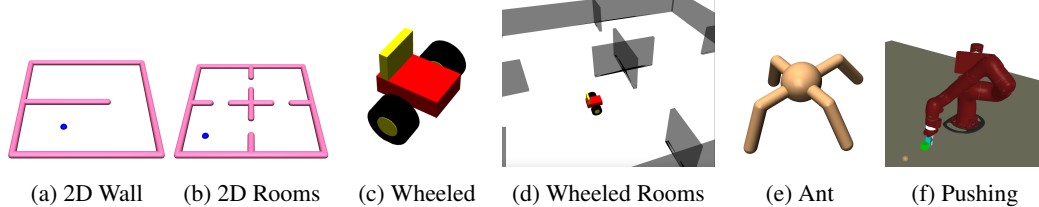

(a) 2D Wall    (b) 2D Rooms    (c) Wheeled    (d) Wheeled Rooms    (e) Ant    (f) Pushing

Figure 4: The tasks in our evaluation. The 2D navigation tasks allow for easy visualization and analysis, while the more complex tasks allow us to investigate how well ARC and prior methods can discern the most functionally-relevant features of the state.

et al., 2015). For a discussion of the assumption about the existence of a goal-conditioned policy, please refer to Section 3. Exact details about the training procedure, the reward function, and hyperparameters are presented in Appendix A.1. To train the ARC representation, we collect a dataset of 500 trajectories with horizon 100 from the goal-conditioned policy, where each trajectory has an arbitrary start state and intended goal state. We optimize Eqn 2 as a supervised learning problem using this dataset to train the representation, computing the relevant expectations by uniform sampling from states in the dataset. A detailed outline of the training procedure, along with hyperparameter and architecture choices, is presented in Appendix A.2.

## 6.3 COMPARISONS WITH PRIOR WORK

We compare ARC to other representation learning methods used in previous works for control: variational autoencoders (Kingma & Welling, 2013) (VAE), variational autoencoders trained for feature slowness (Jonschkowski & Brock, 2015) (slowness), features extracted from a predictive model (Oh et al., 2015) (predictive model), features extracted from inverse models (Agrawal et al., 2016; Burda et al., 2018a), and a naïve baseline that uses the full state space as the representation (state). Details of the exact objectives used to train these methods is provided in Appendix B.

For each downstream task in Section 4, we also compare with alternative approaches for solving the task not involving representation learning. For reward shaping, we compare with VIME (Houthooft et al., 2016), an exploration method based on novelty bonuses. For hierarchical RL, we compare with option critic (Klissarov et al., 2017) and an on-policy adaptation of HIRO (Nachum et al., 2018). We also compare to model-based reinforcement learning with MPC (Nagabandi et al., 2017), a method which explicitly learns and uses environment dynamics, as compared to the implicit dynamics learnt by ARC. Because sample complexity of model-based and model-free methods differ, all results with model-based reinforcement learning indicate final performance.

To ensure a fair comparison between the methods, we provide the *same* information and trajectory data that ARC receives to all of the representation learning methods. Each representation is trained on the same dataset of trajectories collected from the goal-conditioned policy, ensuring that each comparisons receives data from the full state distribution and meaningful transitions.

## 6.4 ANALYSIS OF LEARNED ACTIONABLE REPRESENTATIONS

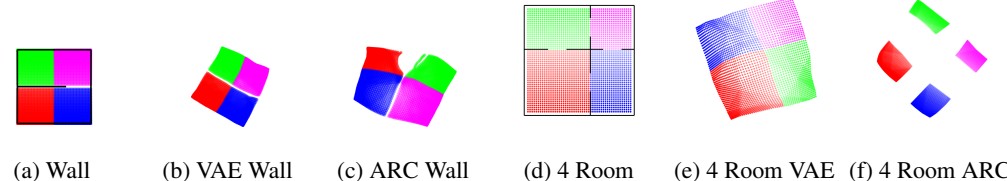

(a) Wall    (b) VAE Wall    (c) ARC Wall    (d) 4 Room    (e) 4 Room VAE    (f) 4 Room ARC

Figure 5: Visualization of ARC for 2D navigation. The states in the environment are colored to help visualize their position in representation space. For the wall task, points on opposite sides of the wall are clearly separated in ARC space (c). For four rooms, we see that ARCs provide a clear decomposition into room clusters (f), while VAEs do not (e).

We analyze the structure of ARC space for the tasks described in Section 6.1, to identify which factors of state ARC chooses to emphasize, and how system dynamics affect the representation.

In the 2D navigation tasks, we visualize the original state and learned representations in Figure 5. In both environments, ARC reflects the dynamics: points close by in Euclidean distance in the original state space are distant in representation space *when they are functionally distinct*. For instance, there is a clear separation in the latent space where the wall should be, and points on opposite sides of the wall are much further apart in ARC space (Figure 5) than in the original environment and in the VAE representation. In the room navigation task, the passages between rooms are clear bottlenecks, and the ARC representation separates the rooms according to these bottlenecks.

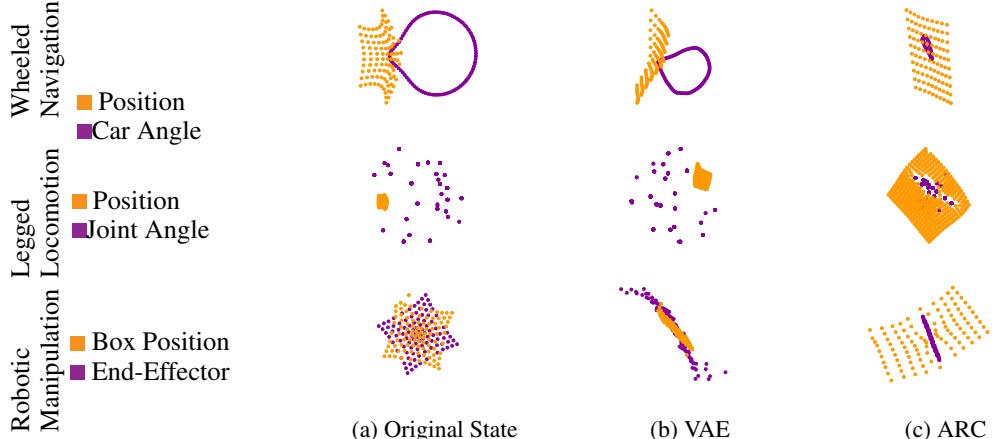

Figure 6: Perturbation analysis (Section 6.4): Effective representations vary significantly with perturbations to functionally relevant elements of state (shown in orange), and less for secondary elements (shown in purple). ARC exhibits this property, with a spread orange region - robot CoM or object position, and a suppressed purple region - joint angles and other secondary elements. The VAE and naive state representations do not capture this saliency, containing spread purple regions.

The representations learned in more complex domains, such as wheeled or legged locomotion and block manipulation, also show meaningful patterns. We aim to understand which elements of state are being emphasized by the representation, by analyzing how distances in the latent space change as we perturb various elements of state. (Fig 6). For each environment, we determine two factors in the state: one which we consider salient for decision making (in orange), and one which is secondary (in purple). We expect a good representation to have a larger variation in distance as we perturb the important factor than when we perturb the secondary factor. In the legged locomotion environment, the CoM is the important factor and the joint angles are secondary. As we perturb the CoM, the representation should vary significantly, while the effect should be muted as we perturb the joints. For the wheeled environment, position of the car should cause large variations while the orientation should be secondary. For the object pushing, we expect block position to be salient and end-effector position to be secondary. Since distances in the high-dimensional representation space are hard to visualize, we project [ARC, VAE, State] representations of perturbed states into 2 dimensions (Fig 6) using multi-dimensional scaling (MDS) (Borg & Groenen, 2005), which projects points while preserving Euclidean distances. From Fig 6, we see that ARC captures the factors of interest; as the important factor is perturbed the representation changes significantly (spread out orange points), while when the secondary factor is perturbed the representation changes minimally (close together purple points). This implies that for Ant, ARC captures CoM while suppressing joint angles; for wheeled, ARC captures position while suppressing orientation; for block pushing, ARC captures block position, suppressing arm movement. Both VAE representations and original state space are unable to capture this.

## 6.5 Leveraging Actionable Representations for Reward Shaping

As desribed in Section 4.2, distances in ARC space can be used for reward shaping to solve tasks that present a large exploration challenge with sparse reward functions. We investigate this on two

challenging exploration tasks for wheeled locomotion and legged locomotion (seen in Fig 7). We acquire an ARC from a goal-conditioned policy in the region $\mathcal{S}$ where the CoM is within a 2m square. The learned representation is then used to guide learning via reward shaping for learning a goal-conditioned policy on a larger region $\mathcal{S}'$, where the CoM is within a square of 8m. The task is to reach arbitrary goals in $\mathcal{S}'$, but with only a *sparse* goal completion reward, so exploration is challenging. We shape the reward with a term corresponding to distance between the representation of the current and desired state: $r(s, g) = r_{\text{sparse}} - \|\phi(s) - \phi(g)\|_2$. To ensure fairness, all comparisons initialize from the goal-conditioned policy on small region $\mathcal{S}$ and train on the same data. Further details on the experimental setup for this domain can be found in Appendix A.3.

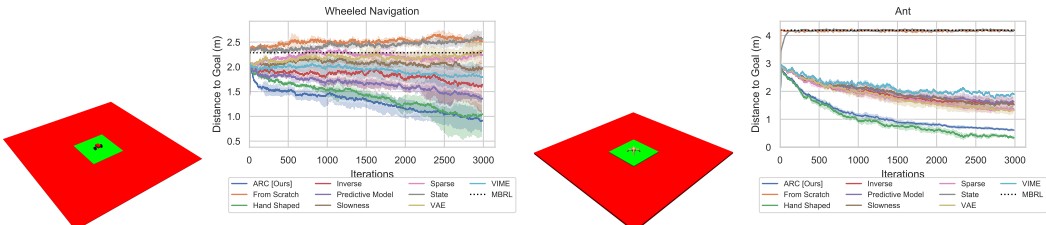

Figure 7: Learning new tasks with reward shaping in representation space. ARC representations are more effective than other methods, and match the performance of a hand-specified shaping.

As shown in Fig 7, ARC demonstrates faster learning speed and better asymptotic performance over all compared methods, when all are initialized from the goal conditioned policy trained on the small region. This can be attributed to the fact that, unlike the other representation learning algorithms, the ARC representation explicitly optimizes for functional distances in latent space, which generalizes well to a larger domain since the functionality in the new space is preserved. The performance of ARC is similar to a hand-designed reward shaping corresponding to distance in COM space, corroborating Figure 6 that ARC considers CoM to be the most salient feature. We notice that representations which are dynamics-aware (ARC, predictive models, inverse models) outperform VIME, which uses a novelty-based exploration strategy without considering environment dynamics, indicating that effectively incorporating dynamics information into representations can help tackle exploration challenges in large environments.

## 6.6 LEVERAGING ACTIONABLE REPRESENTATIONS AS FEATURES FOR LEARNING POLICIES

We consider using the ARC representation as a feature space for learning policies for tasks that cannot be expressed with a goal-reaching objective. We consider a quadruped ant robot task which requires the agent to reach a target (shown in green in Fig 8) *while avoiding* a dangerous region (shown in red in Fig 8). Instead of learning a policy from state $\pi(a|s)$, we learn a policy using a representation $\phi$ as features $\pi(a|\phi(s))$. It is important to note that this task cannot be solved directly by a goal-conditioned policy (GCP), and a GCP attempting to reach the specified goal will walk through the dangerous region and receive a reward of -760. The reward function for this task and other experimental details are noted in Appendix A.4.

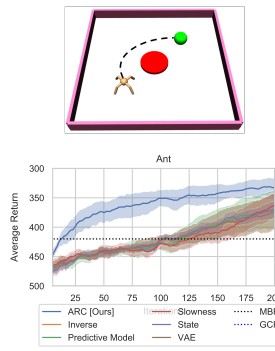

Figure 8: ARCs as policy features. **Top**: Reach-while-avoiding task **Bottom**: Task learning curves

Although all the methods ultimately learn to solve the task, policies using ARC features learn at a significantly faster rate (Figure 8). Policies using ARC features solve the task by Iteration 100, by which point all other methods can only solve with 5% success. We attribute the rapid learning progress to the ability of ARC to emphasize elements of the state that are important for multi-timestep control, rather than greedy features discovered by reconstruction or one-step prediction. Features which emphasize elements important for control make learning easier because they reduce redundancy and noise in the input, and allows the RL algorithm to effectively assign credit. We further note that other representation learning methods learn only as fast as the original state representation, and model-based MPC controllers (Nagabandi et al., 2017) also perform suboptimally. It is important to note that the same representation can be used to quickly train many different tasks, amortizing the cost of training a GCP.

### 6.7 Building Hierarchy from Actionable Representations

We consider using ARC representations to control high-level controllers for learning temporally extended navigation tasks in room and waypoint navigation settings, as described in Section 4. In the multi-room environments, the agent must navigate through a sequence of 50 rooms in order, receiving a sparse reward when it enters the correct room. In waypoint navigation, the ant must reach a sequence of waypoints in order with a similar sparse reward. These tasks are illustrated in Fig 9, and are described in detail in Appendix A.5.

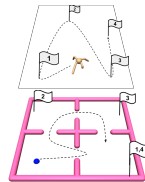

Figure 9: Waypoint and multi-room HRL tasks

We evaluate the two schemes for hierarchical reasoning with ARCs detailed in Section 4.3: commanding directly in representation space or through a $k$-means clustering of the representation space. We train a high-level controller $\pi_h$ with TRPO which outputs as actions either a direct point in the latent space $z_h$ or a cluster index $c_h$, from which a goal $g_h$ is decoded and passed to the goal-conditioned policy to follow for 50 timesteps. Exact specifications and details are in Appendix A.5 and A.6.

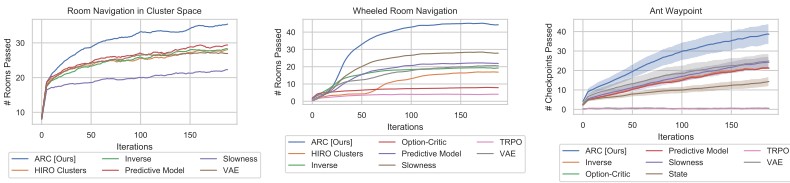

Figure 10: Comparison on hierarchical tasks. ARCs perform significantly better than other representation methods, option-critic, and commanding goals in state space

Using a hierarchical meta-policy with ARCs performs significantly better than those using alternative representations which do not properly capture abstraction and environment dynamics (Fig 10). For multi-rooms, ARC clusters very clearly capture different rooms (Fig 5), so commanding in cluster space reduces redundancy in action space, allowing for effective exploration. ARC likely works better than commanding goals in spaces learned by other representation learning algorithms, because the learned ARC space is more structured for high-level control, which makes search and clustering simpler. Semantically similar states like two points in the same room end up in the same ARC cluster, thus simplifying the high-level planning process for the meta-controller. As compared to learning from scratch via TRPO and standard HRL methods such as option critic (Klissarov et al., 2017) and an on-policy adaptation of HIRO (Nachum et al., 2018), commanding in representation space enables more effective search and high-level control. The failure of TRPO and option-critic, algorithms not using a goal-conditioned policy, emphasizes the task difficulty and indicates that a goal-conditioned policy trained on simple reaching tasks can be re-used to solve long-horizon problems. Commanding in ARC space is better than in state space using HIRO because state space has redundancies which makes search challenging.

## 7 Discussion

In this work, we introduce actionable representations for control (ARC), which capture representations of state important for decision making. We build on the framework of goal-conditioned RL to extract state representations that emphasize features of state that are functionally relevant. The learned state representations are implicitly aware of the dynamics, and capture meaningful distances in representation space. ARCs are useful for tasks such as learning policies, HRL and exploration. While ARC are learned by first training a goal-conditioned policy, learning this policy using off-policy data is a promising direction for future work. Interleaving the process of representation learning and learning of the goal-conditioned policy promises to scale ARC to more general tasks.

**Acknowledgements** This research was supported by Berkeley DeepDrive, Honda, an ONR Young Investigator Program Award, Google, and computational resources from Amazon. Abhishek Gupta was supported by an NSF Graduate Research Fellowship. We thank Pim de Haan, Aviv Tamar, Vitchyr Pong, and Ignasi Clavera for helpful insights and discussions.

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

## A    EXPERIMENTAL DETAILS

### A.1    TRAINING THE GOAL-CONDITIONED POLICY

We train a stochastic goal-conditioned policy $\pi(\cdot|s,g)$ using TRPO with an entropy regularization term, where the goal space $\mathcal{G}$ coincides with the state space $\mathcal{S}$. In every episode, a starting state and a goal state $s, g \in \mathcal{S}$ are sampled from a uniform distribution on states, with a sparse reward given of the form below, where $\epsilon$ is task-specific, and listed in the table below.

$$r(s,g) = \begin{cases} 0 & \|s-g\|_\infty > \epsilon \\ \epsilon - \|s-g\|_\infty & \|s-g\|_\infty < \epsilon \end{cases}$$

For the Sawyer environment, although this sparse reward formulation can learn a goal-conditioned policy, it is highly sample inefficient, so in practice we use a shaped reward as detailed in Appendix C. For all of the other environments, in the free space and rooms environments, the goal-conditioned policy is trained using a sparse reward.

The goal-conditioned policy is parameterized as $\pi_\theta(a|s,g) \sim \mathcal{N}(\mu_\theta(s,g), \Sigma_\theta)$. The mean, $\mu_\theta(\cdot, \cdot)$ is a fully-connected neural network which takes in the state and the desired goal state as a concatenated vector, and has three hidden layers containing 150, 100, and 50 units respectively. $\Sigma$ is a learned diagonal covariance matrix, and is initially set to $\Sigma = I$.

|  | Navigation | Wheeled Navigation | Ant Navigation | Sawyer Pushing |
|---|---|---|---|---|
| State/Goal Space Dimension | 2 | 6 | 15 | 6 |
| Action Space Dimension | 2 | 2 | 7 | 3 |
| Sparse Reward Threshold ($\epsilon$) | 0.1 | 0.5 | 1 | 0.3 |
| # Trajectories per Iteration | 100 | 200 | 250 | 500 |
| # Steps in Trajecotry | 50 | 100 | 200 | 100 |
| # Iterations | 200 | 1000 | 2000 | 2000 |
| Entropy Penalty | 1 | 1 | 0.1 | 0.1 |
| Learning Rate | .01 | .02 | .02 | .01 |

### A.2    TRAINING THE REPRESENTATION

After training a goal-conditioned policy $\pi$ on the specified region of interest, we collect 500 trajectories each of length 100 timesteps, where each trajectory starts at an arbitrary start state, going towards an arbitrary goal state, selected exactly as the goal-conditioned policy was trained in Appendix A.1. This dataset was chosen to be large enough so that the collected dataset has full coverage of the entire state space. Each of the representation learning methods evaluated is trained on this dataset, which means that each learning algorithm receives data from the full state space, and witnesses meaningful transitions between states $(s_t, s_{t+1})$.

We evaluate ARCs against representations minimizing reconstruction error (VAE, slowness) and representations performing one-step prediction (predictive model, inverse dynamics). For each representation, each component is parametrized by a neural network with ReLU activations and linear outputs, and the objective function is optimized using Adam with a learning rate of $10^{-3}$, holding out 20% of the trajectories as a validation set. We perform coarse hyperparameter sweeps over various hyperparameters for all of the methods, including the dimensionality of the latent state, the size of the neural networks, and the parameters which weigh the various terms in the objectives. The exact objective functions for each representation are detailed further in Appendix B.

### A.3    REWARD SHAPING

We test the reward shaping capabilities of the learned representations with a set of navigation tasks on the `Wheeled` and `Ant` tasks. A goal-conditioned policy $\pi$ is trained on a $n \times n$ meter square of free space, and representations are learned (as specified above) on trajectories collected in this small region. We then attempt to generalize to an $m \times m$ meter square (where $m >> n$), and consider the set of tasks of reaching an arbitrary goal in the larger region: a start state and goal state are chosen

uniformly at random every episode. The environment setup is identical to that in Appendix A.1, although with a larger region, and policy training is done the same with two distinctions. Instead of training with the sparse reward $r_{sparse}(s, g)$, we train on a "shaped" surrogate reward

$$r_{shaped,\phi}(s, g) = r_{sparse}(s, g) - \alpha \|\phi(s) - \phi(g)\|_2$$

where $\alpha$ weights between the euclidean distance and the sparse reward terms. Second, the policy is initialized to the parameters of the original goal-conditioned policy $\pi$ which was previously trained on the small region to help exploration. As a heuristic for the best possible shaping term, we compare with a "hand-specified" In addition to reward shaping with the various representations, we also compare to a dedicated exploration algorithm, VIME (Houthooft et al., 2016), which also uses TRPO as a base algorithm. Understanding that different representation learning methods may learn representations with varying scales, we performed a hyperparameter sweep on $\alpha$ for all the representation methods. For VIME, we performed a hyperparameter sweep on $\eta$. The parameters used for TRPO are exactly those in Appendix A.1, albeit for 3000 iterations.

## A.4 FEATURES FOR POLICIES

We test the ability of the representation to be used as features for a policy learning some downstream task within the `Ant` environment. The downstream task is a "reach-while-avoid" task, in which the Ant requires the quadruped robot to start at the point $(-1.5, -1.5)$ and reach the point $(1.5, 1.5)$ while avoiding a circular region centered at the origin with radius 1 (all units in meters). Letting $d_{goal}(s)$ be the distance of the agent to $(1.5, 1.5)$ and $d_{origin}(s)$ to be the distance of the agent to the origin, the reward function for the task is

$$r(s) = -d_{goal}(s) - 4 * \mathbf{1}\{d_{origin}(s) < 1\}$$

For any given representation $\phi(s)$, we train a policy which uses the feature representation as input as follows. We use TRPO to train a stochastic policy $\pi(a|\phi(s))$, which is of the form $\mathcal{N}(\mu_\theta(\phi(s)), \Sigma_\theta)$. The mean is a fully connected neural network which takes in the representation, and has two layers of size 50 each, and $\Sigma$ is a learned diagonal covariance matrix initially set to $\Sigma = I$. Note that gradients do not flow through the representation, so only the policy is adapted and the representation is fixed for the entirety of the experiment.

## A.5 HIERARCHICAL REINFORCEMENT LEARNING IN LATENT SPACE

We provide comparisons on using the learned representation to direct a goal-conditioned policy for long-horizon sequential tasks. In particular, we consider a waypoint reaching task for the Ant, in which the agent must navigate to a sequence of 50 target locations in order: $\{(x_1, y_1), (x_2, y_2), \ldots (x_{50}, y_{50})\}$. The agent receives as input the state of the ant and the checkpoint number that it is currently trying to reach (encoded as a one-hot vector). When the agent gets within 0.5m of the checkpoint, it receives $+1$ reward, and the checkpoint is moved to the next point, making this a highly sparse reward. Target locations are sampled uniformly at random from a $8 \times 8$ meter region, but are fixed for the entirety of the experiment.

We consider learning a high-level policy $\pi_h(z_h|s)$ which outputs goals in latent space, which are then executed by a goal-conditioned policy as described in Appendix A.1. Specifically, when the high-level policy outputs a goal in latent space $z_h$, we use a reconstruction network $\psi$, which is described below, to receive a goal state $g_h = \psi(z_h)$. The goal-conditioned policy executes for 50 timesteps according to $\pi(a|s, g_h)$. The high-level policy is trained with TRPO with the reward being equal to the sum of the rewards obtained by running the goal-conditioned policy for every meta-step. We parametrize the high-level policy $\pi_h(z_h|s)$ as having a Gaussian distribution in the latent space, with the mean being specified as a MLP with two layers of 50 units and Tanh activations, and the covariance as a learned diagonal matrix independent of state.

To allow the latent representation $z$ to provide commands for the goal-conditioned policy, we separately train a reconstruction network $\psi$ which minimizes the loss function $\mathbb{E}_s[\|\psi(\phi(s)) - s\|_2]$. For any latent $z$, we can now use $\psi(z)$ as an input into the goal-conditioned policy. Note that an alternative method of providing commands in latent space is to train a new goal-conditioned policy

$\pi_\phi$, which is trained to minimize the loss $\mathbb{E}_{s,g}[D_{KL}(\pi_\phi(\cdot|s, \phi(g))\|\pi(\cdot|s, g))]$, however to maintain abstraction between the representation and the goal-conditioned policy, we choose the former approach.

## A.6 HIERARCHICAL REINFORCEMENT LEARNING IN CLUSTER SPACE

We provide comparisons on using the learned representation to direct a goal-conditioned policy in **cluster space**, as described in Section 4. We consider navigation through *a sequence of rooms in order* in the rooms and wheeled rooms environment, as visualized in Figure 4. A sequence of 50 checkpoints are sampled uniformly from the four rooms with the extra constraint that the same room is never repeated two checkpoints in a row (that is, each checkpoint is chosen to be any of the four rooms), and held fixed for the entirety of the experiment. The agent is tasked with going through these rooms in order, receiving a $+1$ reward every time it enters the appropriate room. The policy receives as input the state of the agent, and which number checkpoint the agent is currently trying to reach (encoded as a 50-dimensional one-hot vector).

After having learned a representation $\phi$ using some set of trajectory data, as described in Appendix A.2, we run $k$-means clustering on states in the trajectory data to cluster latent states in the representation into $k$ components. We then consider learning a high-level policy $\pi_h(c_h|s)$ which outputs a cluster between $\{1 \ldots k\}$. Given a cluster number $c_h$ from the high-level policy, the low-level policy samples a latent state $z_h$ uniformly from the cluster, and then proceeds to command a learnt goal-conditioned policy exactly as described in Appendix A.5.

Specifically, we learn a high-level policy of the form $\pi_h(c_h|s) \sim \text{Categorical}(p_\theta(s))$ using TRPO where the probabilities for each cluster are specified by a neural network $\pi_\theta$ which has two layers of 50 units each, with Tanh activations, and a final Softmax activation to normalize outputs into the probability simplex.

We performed hyperparameter sweeps over $k$ - the number of clusters - for each representation method.

## B  BENCHMARK REPRESENTATIONS

We provide the loss functions that are used to train each of the representations evaluated in our work. All representations are trained on a dataset of trajectories $\mathcal{D} = \{\tau_i\}_{i=1}^n$. We use the notation $s \sim \mathcal{D}$ to denote sampling a state uniformly at random from a trajectory uniformly at random from the dataset. We use the notation $s_t, s_{t+1} \sim \mathcal{D}$ to denote sampling a state and the state right after it according to the same uniform sampling scheme.

- **ARC** - After precomputing $D_{act}$:a matrix of actionable distances, we train a neural network $\phi$ to minimize

$$D_{act}(s_i, s_j) = \mathbb{E}_{s \sim \mathcal{D}}\left[D_{KL}(\pi(a|s, s_i)\|\pi(a|s, s_j))\right]$$

$$\mathcal{L}(\phi) = \mathbb{E}_{s \sim \mathcal{D}}\left[\mathbb{E}_{s' \sim \mathcal{D}}\left[|\|\phi(s) - \phi(s')\| - D_{act}(s, s')\|^2\right]\right]$$

- **VAE** (Kingma & Welling, 2013) - Given $q_\phi(z|x) = \mathcal{N}(\mu_\phi(x), \sigma_\phi(x))$, $p_\theta(x|z) = \mathcal{N}(\psi_\theta(z), I)$, and $p(z) = \mathcal{N}(0, I)$

$$\mathcal{L}(\phi, \theta) = \mathbb{E}_{s \sim \mathcal{D}}\left[\mathbb{E}_{z \sim q_\phi(x)}\left[\log p_\theta(x|z) - \beta D_{KL}(q_\theta(z|x)\|p(z))\right]\right]$$

  Here $\mu_\phi, \sigma_\phi, \psi_\theta$ are all neural networks, and $\beta$ is a tunable hyperparameter. The log-likelihood term is equivalent to minimizing mean squared error.

- **Slowness** (Jonschkowski & Brock, 2015) - Given $q_\phi(z|x) = \mathcal{N}(\mu_\phi(x), \sigma_\phi(x))$, $p_\theta(x|z) = \mathcal{N}(\psi_\theta(z), I)$, and $p(z) = \mathcal{N}(0, I)$

$$\mathcal{L}(\phi, \theta) = \mathbb{E}_{(s_t, s_{t+1}) \sim \mathcal{D}}\left[\mathbb{E}_{z \sim q_\phi(s_t)}\left[\log p_\theta(s_t|z) - \beta D_{KL}(q_\theta(z|x)\|p(z)) - \alpha\|\mu_\theta(s_{t+1} - \mu_\theta(s_t)\|\right]\right]$$

  Here $\mu_\phi, \sigma_\phi, \psi_\theta$ are all neural networks, and $\alpha, \beta$ are tunable hyperparameters. The log-likelihood terms are equivalent to minimizing mean squared error.

- **Predictive Model** (Oh et al., 2015) - Given $z = \phi(s_t)$, $\hat{z}_{t+1} = f(z, a)$ and $\psi(z) = \hat{s}$, we

$$\mathcal{L}(\phi, f, \psi) = \mathbb{E}_{(s_t, a_t, s_{t+1}) \sim \mathcal{D}} \left[ \|s_{t+1} - \psi(f(\phi(s_t), a_t))\|_2^2 \right]$$

where $\phi$ is the learnt representation, $f$ a model in representation space, and $\psi$ a reconstruction network retrieving are all neural networks.

- **Inverse Model** (Burda et al., 2018a) - Given $z_t = \phi(s_t)$, $\hat{z}_{t+1} = f(z_t, a)$, $\hat{a}_{t+1} = g(z_t, z_{t+1})$

$$\mathcal{L}(\phi, f, g) = \mathbb{E}_{(s_t, a_t, a_{t+1}) \sim \mathcal{D}} \left[ \|a_t - g(\phi(s_t), \phi(s_{t+1}))\|_2^2 + \beta \|\phi(s_{t+1}) - f(\phi(s_t), a_t)\|_2^2 \right]$$

Here, $\phi$ is the learnt representation, $f$ is a learnt model in the representation space, and $g$ is a learnt inverse dynamics model in the representation space. $\beta$ is a hyperparameter which controls how forward prediction error is balanced with inverse prediction error.

## C  TASK DESCRIPTIONS

- **2D Navigation** This environment consists of an agent navigating to points in an environment, either with a wall as in Figure 4a or with four rooms, as in Figure 4b. The state space is 2-dimensional, consisting of the Cartesian coordinates of the agent. The agent has acceleration control, so the action space is 2-dimensional. Downstream tasks for this environment include reaching target locations in the environment and navigating through a sequence of 50 rooms.

- **Wheeled Navigation** This environment consists of a car navigating to locations in an empty region, or with four rooms, as illustrated in Figure 4. The state space is 6-dimensional, consisting of the Cartesian coordinates, heading, forward velocity, and angular velocity of the car. The agent controls the velocity of both of its wheels, resulting in a 2-dimensional action space. Goal-conditioned policies are trained within a $3 \times 3$ meter square.

  Downstream tasks for wheeled navigation include reaching target locations in the environment, navigating through sequences of rooms, and navigating through sequences of waypoints.

- **Ant** This task requires a quadrupedal ant robot navigating in free space. The state space is 15-dimensional, consisting of the Cartesian coordinates of the ant, body orientation as a quaternion, and all the joint angles of the ant. The agent must use torque control to control it's joints, resulting in an 8-dimensional action space. Goal conditioned policies are trained within a $2 \times 2$ meter square.

  Downstream tasks for the ant include reaching target locations in the environment, navigating through sequences of waypoints, and reaching target locations while avoiding other locations.

- **Sawyer** This environment involves a Sawyer manipulator and a freely moving block on a table-top. The state space is 6-dimensional, consisting of the Cartesian coordinates of the end-effector of the Sawyer, and the Cartesian coordinates of the block. The Sawyer is controlled via end-effector position control with a 3-dimensional action space.

  Because training a goal-conditioned policy takes an inordinate number of samples for the Sawyer environment, we instead use the following shaped reward to train the GCVF where $h(s)$ is the position of the hand and $o(s)$ is the position of the object

$$r_{shaped}(s, g) = r_{sparse}(s, g) - \|h(s) - o(s)\| - 2\|o(s) - o(g)\|$$

## D  HYPERPARAMETER TUNING

We perform hyperparameter tuning on three ends: one to discover appropriate parameters for each representation for each environment which are then held constant for the experimental analysis, then on the downstream applications, to choose a scaling factor for reward shaping (see Appendix A.3), and to choose the number of clusters for the hierarchical RL experiments in cluster space (see Appendix A.6).

To discover appropriate parameters for each representation for the legged and wheeled locomotion environments, we evaluate representations on the downstream reward-shaping task, performing a hyperparameter sweep on latent dimension and the parameters which weigh the various terms in the representation learning objectives. We keep the network architecture fixed for each representation and each task. We emphasize carefully here that the ARC representation requires *no parameters* to tune beyond the size of the latent dimension, and we perform a hyperparameter sweep on the penalty terms to ensure that other methods aren't improperly penalized. On the size of the latent dimension, we sweep over $\{2, 3, 4\}$ for wheeled locomotion and $\{3, 5, 7, 9, 11\}$ for the ant. For the relative weighting for the penalty terms for the comparison representations (defined by $\beta$ in Appendix B), we evaluate possible values $\beta \in \{4^{-2}, 4^{-1}, 1, 4^1, 4^2\}$. These representations are then fixed and used for all the downstream applications.

For reward shaping, we tune the relative scales between the sparse reward and the shaping term, (denoted by $\alpha$ in Appendix A.3) over possible values $\alpha \in \{1, 4^1, 4^2, 4^3, 4^4\}$ for each representation on both the legged and wheeled locomotion environments. Tuning for $\alpha$ is required because the representations may have different latent dimensions and different scales, and chose to perform this hyperparameter sweep instead of adding a term to the representation learning objectives to ensure uniformity in scale. For performing $k$-means clustering on the HRL cluster experiments, we sweep over possible values $k \in \{4, 5, 6, 7, 8\}$ for each representation on the room navigation tasks for 2D and wheeled navigation, but however found that most representations were robust to choice of the number of clusters.

