# OpenReview forum: "Learning Actionable Representations with Goal Conditioned Policies"
_ICLR.cc/2019/Conference_

### Official Review · AnonReviewer3 · 2018-10-16
**Quite interesting idea, but unsufficiently mature piece of research**

**Rating:** 6
**Confidence:** 4

**Review:**

In this paper, the authors propose a new approach to representation learning in the context of reinforcement learning.
The main idea is that two states should be distinguished *functionally* in terms of the actions that are needed to reach them,
in contrast with generative methods which try to capture all aspects of the state dynamics, even those which are not relevant for the task at hand.
The method of the authors assumes that a goal-conditioned policy is already learned, and they use a Kullback-Leibler-based distance
between policies conditioned by these two states as the loss that the representation learning algorithm should minimize.
The experimental study is based on 6 simulated environments and outlines various properties of the framework.

Overall, the idea is interesting, but the paper suffers from many weaknesses both in the framework description and in the experimental study that make me consider that it is not ready for publication at a good conference like ICLR.

The first weakness of the approach is that it assumes that a learned goal-conditioned policy is already available, and that the representation extracted from it can only be useful for learning "downstream tasks" in a second step. But learning the goal-conditioned policy from the raw input representation in the first place might be the most difficult task. In that respect, wouldn't it be possible to *simultaneously* learn a goal-conditioned policy and the representation it is based on? This is partly suggested when the authors mention that the representation could be learned from only a partial goal-conditioned policy, but this idea definitely needs to be investigated further.

A second point is about unsufficiently clear thoughts about the way to intuitively advocate for the approach. The authors first claim that two states are functionally different if they are reached from different actions. Thinking further about what "functionally" means, I would rather have said that two states are functionally different if different goals can be reached from them. But when looking at the framework, this is close to what the authors do in practice: they use a distance between two *goal*-conditioned policies, not *state*-conditioned policies. To me, the authors have established their framework thinking of the case where the state space and the goal space are identical (as they can condition the goal-conditioned policy by any state=goal). But thinking further to the case where goals and states are different (or at least goals are only a subset of states), probably they would end-up with a different intuitive presentation of their framework. Shouldn't finally D_{act} be a distance between goals rather than between states?

Section 4 lists the properties that can be expected from the framework. To me, the last paragraph of Section 4 should be a subsection 4.4 with a title such as "state abstraction (or clustering?) from actionable representation". And the corresponding properties should come with their own questions and subsection in the experimental study (more about this below).

About the related work, a few remarks:
- The authors do not refer to papers about using auxiliary tasks. Though the purpose of these works is often to supply for additional reward signals in the sparse reward context, then are often concerned with learning efficient representations such as predictive ones.
- The authors refer to Pathak et al. (2017), but not to the more recent Burda et al. (2018) (Large-scale study of curiosity-driven learning) which insists on the idea of inverse dynamical features which is exactly the approach the authors may want to contrast theirs with. To me, they must read it.
- The authors should also read Laversanne-Finot et al. (2018, CoRL) who learn goal space representations and show an ability to extract independently controllable features from that.

A positive side of the experimental study is that the 6 simulated environments are well-chosen, as they illustrate various aspects of what it means to learn an adequate representation. Also, the results described in Fig. 5 are interesting. A side note is that the authors address in this Figure a problem pointed in Penedones et al (2018) about "The Leakage Propagation problem" and that their solution seems more convincing than in the original paper, maybe they should have a look.
But there are also several weaknesses:
- for all experiments, the way to obtain a goal-conditioned policy in the first place is not described. This definitely hampers reproducibility of the work. A study of the effect of various optimization effort on these goal-conditioned policies might also be of interest.
- most importantly, in Section 6.4, 6.5 and 6.6, much too few details are given. Particularly in 6.6, the task is hardly described with a few words. The message a reader can get from this section is not much more than "we are doing something that works, believe us!". So the authors should choose between two options:
* either giving less experimental results, but describing them accurately enough so that other people can try to reproduce them, and analyzing them so that people can extract something more interesting than "with their tuning (which is not described), the framework of the authors outperforms other systems whose tuning is not described either".
* or add a huge appendix with all the missing details.
I'm clearly in favor of the first option.

Some more detailed points or questions about the experimental section:
- not so important, Section 6.2 could be grouped with Section 6.1, or the various competing methods could be described directly in the sections where they are used.
- in Fig. 5, in the four room environment, ARC gets 4 separated clusters. How can the system know that transitions between these clusters are possible?
- in Section 6.3, about the pushing experiment, I would like to argue against the fact that the block position is the important factor and the end-effector position is secundary. Indeed, the end-effector must be correctly positioned so that the block can move. Does ARC capture this important constraint?
- Globally, although it is interesting, Fig.6 only conveys a quite indirect message about the quality of the learned representation.
- Still in Fig. 6, what is described as "blue" appears as violet in the figures and pink in the caption, this does not help when reading for the first time.
- In Section 6.4, Fig.7 a, ARC happens to do better than the oracle. The authors should describe the oracle in more details and discuss why it does not provide a "perfect" representation.
- Still in Section 6.4, the authors insist that ARC outperforms VIME, but from Fig.7, VIME is not among the best performing methods. Why insist on this one? And a deeper discussion of the performance of each method would be much more valuable than just showing these curves.
- Section 6.5 is so short that I do not find it useful at all.
- Section 6.6 should be split into the HRL question and the clustering question, as mentioned above. But this only makes sense if the experiments are properly described, as is it is not useful.

Finally, the discussion is rather empty, and would be much more interesting if the experiments had been analyzed in more details.

typos:

p1: that can knows => know
p7: euclidean => Euclidean

---

> ### Author Response · Authors · 2018-11-17
> **Response to Reviewer 3 (Continued)**
>
> Find responses to particular comments below:
> Related work:
> -> We cite and discuss all the papers mentioned in the related work section (Section 5). We additionally added comparison (Fig 7,8,10) to using inverse dynamics models and model-based RL methods, as discussed above.
>
> “Shouldn't finally D_{act} be a distance between goals rather than between states?”
> > D_{act} is indeed the actionable distance between goals, but given that the goal and the state space are the same the learned representation can be effectively used as a state representation as seen in Section 6.6.
>
> “in Fig. 5, in the four room environment, ARC gets 4 separated clusters. How can the system know that transitions between these clusters are possible?”
> ->  We have added a discussion in Section 6.6 to clarify this. We use model free RL to train the high level policy which directly outputs clusters as described in Section 4.4. This high level policy does not need to explicitly model the transitions between clusters, that is handled by the low level goal reaching policy, and the high-level policy is trained model-free.
>
> “Indeed, the end-effector must be correctly positioned so that the block can move. Does ARC capture this important constraint?”
> -> ARC does not completely ignore the end effector position, this is evidenced from the fact that the blue region in Fig 6 is not a point but is an entire area. What ARC captures is that moving the block induces a greater difference in actions than inducing the arm. Moving the block to different positions requires the arm to move to touch the block and push it to the goal, while moving the arm to different positions can be done by directly moving it to the desired position. While both things are captured, the block is emphasized over the end-effector.
>
> “In Section 6.4, Fig.7 a, ARC happens to do better than the oracle. why?”
> -> The oracle comparison is a hand-specified reward shaping - we have updated Section 6.5 and Figure 7 to make this point clear. It is likely that the ARC representation is able to find an even better reward shaping, although the difference is fairly small.
>
> “from Fig.7, VIME is not among the best performing methods. Why insist on this one?”
> -> We intended to emphasize that ARC is able to outperform a method that is purely designed for better exploration, not just other methods for representation learning. The discussion in Section 6.5 has been appropriately altered.
>
> [1] Nair, Pong, Dalal, Bahl, Lin, and Levine. Visual reinforcement learning with imagined goals.NIPS 2018
> [2] Pong, Gu, Dalal, and Levine. Temporal difference models: Model-free deep rl for model-based control. ICLR 2018
> [3] Andrychowicz, Wolski, Ray, Schneider, Fong, Welinder, P., McGrew, B., Tobin, J., Abbeel, P., and Zaremba, W. (2017). NIPS 2017
> [4] Burda, Edwards, Pathak, Storkey, Darrell, and Efros. Large-scale study of curiosity-driven learning. arXiv preprint
> [5] Nagabandi, Kahn, Fearing and Levine. Neural Network Dynamics for Model-Based Deep Reinforcement Learning with Model-Free Fine-Tuning. ICRA 2018

---

> > ### Comment · AnonReviewer3 · 2018-11-22
> > **Serious improvement, but some points are remaining**
> >
> > The authors have done a large effort in addressing a lot of our concerns, particularly regarding experimental details, how to learn goal conditioned policies (GCPs), and the related work section has been improved. The paper is now better and I will increase my score accordingly when appropriate.
> >
> > However, the fact that GCPs need to be learned in advance before the ARC representation can be learned still raises a major concern that needs further clarification, probably with some impact on the introduction and the positioning of the paper.
> >
> > The question is the following: if GCPs are learned and authors considers it is rather "easy" to do so, why do we need something more? If you take the title of Sections 4.1 and 6.6, why "leveraging actionable representation as feature for learning policies" if you already learned policies ? The last paragraph suggests that policies learned in ARC space will generalize "beyond GCPs". Since GCPs limitations have not been made clear, this point is still vague.
> >
> > In Section 4, the authors suggest three other valuable answers to this question: reward shaping, doing HRL, or clustering in ARC space (by the way, the latter could be used to help the former). My feeling is that treating those three points in addition to the one above is somewhat dispersive, the paper is trying to make to many points, at least a unifying perspective is missing.
> >
> > To me, the paper lacks between Section 4 and Section 4.1 an introductiory text which should contain the last paragraph of Section 3 and would motivate the work more clearly with respect to the above issue.
> >
> > If the paper does not get finally accepted at ICLR, I would suggest the authors to reconsider their positionning with respect to the perspective above and put forward a clearer message about what actionable representations really bring in a context where you already have "good enough" GCPs.
> >
> > Finally, the perspective mentioned in the discussion of interleaving ARC learning and GCP learning would of course change the picture about the above issue, I appreciate that the authors kept that for their last sentence.

---

> > > ### Author Response · Authors · 2018-11-24
> > > **Response to Reviewer 3: Regarding the Goal-Conditioned Policy**
> > >
> > > Thank you for your response and helpful suggestions! The question you raise about the necessity of something beyond a goal-conditioned policy is a valuable one, and we answer it below. We have updated the discussion in Section 4 to reflect the same. We have also added an additional comparison to directly using goal conditioned policies in Section 6.6.
> > >
> > > Although GCPs trained to reach one state from another are feasible to learn, they possess some fundamental limitations (added discussion in Section 4): they do not generalize very well to new states, and they are limited to solving tasks expressible as just reaching a particular goal.  The unifying explanation for why ARCs are useful over just a GCP is that the learned representation generalizes better than the GCP - to new tasks and to new regions of the environment. In our experimental evaluation, we show that ARCs can help solve tasks that cannot be expressed as goal reaching (Section 6.6, 6.7) and they enable learning policies on larger regions to which GCPs do not generalize (Section 6.5).
> > >
> > > As the GCP is trained with a sparse reaching reward, it is unaware of possible reward structures in the environment, making it hard to adapt to tasks which are not simple goal reaching, such as the “reach-while-avoid” task in Section 6.6. For this task, following the GCP directly would cause the ant to walk through the red region and incur a large negative reward; a comparison we now explicitly add to Section 6.6. Tasks which cannot be expressed as simply reaching a goal are abundant in real life scenarios such as navigation with preferences or manipulation with costs on quality of motion, and fast learning on such tasks (as ARC does) is quite beneficial. We have explicitly emphasized this discussion in Section 4.1, and made the limitations of simple goal reaching clear at the start of Section 4.
> > >
> > > In the tasks for Section 6.5, the GCP trained on the 2m region (in green) does not achieve high performance on the larger region of 8m, even when finetuned on the environment using the provided reward (Fig 8). However, shaping the reward function using ARCs enables learning beyond the the original GCP, showing that ARCs generalize better to this new region, and potentially can lead to learning progressively harder GCPs via bootstrapping.
> > >
> > > We agree that the discussion would greatly benefit from an introductory paragraph putting things into context, we have added this discussion at the beginning of Section 4. Please let us know if this resolves the issues you brought up. If not, we’re happy to address any other concerns you might have.

---

> ### Author Response · Authors · 2018-11-17
> **Response to Reviewer 3**
>
> Thank you for your insightful comments and suggestions! We have made many changes based on the comments provided by reviewers, which are summarized below. We would appreciate it if the reviewer could take another look at our changes and additional results, and let us know if they would like to revise their score or request additional changes that would alleviate their concerns.
>
> New comparisons:
> We have added two more comparisons as suggested - with model based RL methods ([5] Nagabandi et al) and learning representations via inverse dynamics models ([4] Burda et al). These have been described in Section 6.3 and added to plots in Fig 7, 8, 10. We have also added a new comparison to learning from scratch for the reward shaping experiment (Section 6.5, Fig 7).
>
> Lack of details:
> We apologize for the lack of clarity in the submission! We have updated the main text and added an appendix with additional details of the ARC representation and the experimental setup: how a goal-conditioned policy is trained (Sec 6.2, Appendix A.1), how the ARC representation is learned (Sec 6.2, Appendix A.2) , and how the methods are evaluated on downstream applications (Sec 6.5-7, Appendix A.3-6). We increased analysis of the performance of ARC and comparison methods for all the downstream applications (Sec 6.5-6.7), and added a discussion of how all methods are trained (Sec 6.3, Appendix A.2, B)
>
> Requirement for goal-conditioned policy:
> The ARC representation is extracted from a goal-conditioned policy (GCP), requiring us to assume that we can train such a GCP. This assumption was explicit in our submission, but we have emphasized it more now by editing Section 1 and Section 3. For our experiments, the GCP was trained with existing RL methods using a sparse task-agnostic reward (Section 6.2, Appendix A.1) -- obtaining such a policy is not especially difficult, and existing methods are quite capable of doing so [1,2,3].  We therefore believe that this assumption is reasonable. We also ensure that our experiments fairly account for the data required to train the GCP.
> - In the generalization experiment (Section 6.4), all methods initialize behaviour from the GCP, as policies trained from scratch fail, a new comparison we have added to Figure 7.
> - In the hierarchy experiment (Section 6.7), all representations use the GCP as a low-level controller. Two comparisons (TRPO, Option Critic) which do not use the GCP make zero progress, even when provided with substantially more samples.
> - In learning non goal-reaching tasks (Section 6.6), ARC representation can be re-used across many tasks without retraining the GCP, amortizing the cost of learning the GCP.  We plan to add an experimental comparison on a family of tasks to demonstrate this, and will update the paper.

---

### Official Review · AnonReviewer2 · 2018-10-31
**Paper lacks many important details.**

**Rating:** 6
**Confidence:** 4

**Review:**

The paper presents a method to learn representations where proximity in euclidean distance represents states that are achieved by similar policies. The idea is novel (to the best of my knowledge), interesting and the experiments seem promising. The two main flaws in the paper are the lack of details and missing important experimental comparisons.

Major remarks:

- The author state they add experimental details and videos via a link to a website. I think doing so is very problematic, as the website can be changed after the deadline but there was no real information on the website so it wasn’t a problem this time.

- While the idea seems very interesting, it is only presented in very high-level. I am very skeptical someone will be able to reproduce these results based only on the given details. For example - in eq.1 what is the distribution over s? How is the distance approximated? How is the goal-conditional policy trained? How many clusters and what clustering algorithm?

- Main missing details is about how the goal reaching policy is trained. The authors admit that having one is “a significant assumption” and state that they will discuss why it is reasonable assumption but I didn’t find any such discussion  (only a sentence in 6.4).

- While the algorithm compare to a variety of representation learning alternatives, it seems like the more natural comparison are model-based Rl algorithms, e.g. “Neural Network Dynamics for Model-Based Deep Reinforcement Learning with Model-Free Fine-Tuning”. This is because the representation tries to implicitly learn the dynamics so it should be compared to models who explicitly learn the dynamics.

- As the goal-conditional policy is quite similar to the original task of navigation, it is important to know for how long it was trained and taken into account.

- I found Fig.6 very interesting and useful, very nice visual help.

- In fig.8 your algorithm seems to flatline while the state keeps rising. It is not clear if the end results is the same, meaning you just learn faster, or does the state reach a better final policy. Should run and show on a longer horizon.

---

> ### Author Response · Authors · 2018-11-17
> **Response to Reviewer 2 (Continued)**
>
> Find responses to particular questions and comments below:
> “Should run and show on a longer horizon.”
> -> We have updated Figure 8 accordingly. All methods converge to the same average reward.
>
> “As the goal-conditional policy is quite similar to the original task of navigation, it is important to know for how long it was trained and taken into account.”
> -> We have added these details in Appendix A.1. It is important to note that for the task in Section 6.6, simply using a goal reaching policy would be unable to solve the task, since it has no notion of other rewards, like regions to avoid (shown in red in Fig 8), and would pass straight through the region.
>
> “eq.1 what is the distribution over s?”
> -> It is the distribution over all states over which the goal-conditioned policy is trained. This is done by choosing uniformly from states on trajectories collected with the goal-conditioned policy as described in Section 6.2 and Appendix A.2.
>
> “ How is the distance approximated?”
> -> In our experimental setup, we parametrize the action distributions of GCPs with Gaussian distributions - for this class of distributions, the KL divergence, and thus the actionable distance, can be explicitly computed (Appendix A.1).
>
> “How many clusters and what clustering algorithm?”
> -> We use k-means for clustering, with distance in ARC space as the metric. We perform a hyperparameter sweep over the number of clusters for each method, and thus varies across tasks and methods. We have added this clarification to Section 4.4 and Section 6.6.
>
>
> The author state they add experimental details and videos via a link to a website.
> > OpenReview does not provide a mechanism for submitting supplementary materials. Providing supplementary materials via an external link is the instruction provided by the conference organizers -- we would encourage the reviewer to check with the AC if they are concerned.
>
> [1] Nair, Pong, Dalal, Bahl, Lin, and Levine. Visual reinforcement learning with imagined goals.NIPS 2018
> [2] Pong, Gu, Dalal, and Levine. Temporal difference models: Model-free deep rl for model-based control. ICLR 2018
> [3] Andrychowicz, Wolski, Ray, Schneider, Fong, Welinder, P., McGrew, B., Tobin, J., Abbeel, P., and Zaremba, W. (2017). NIPS 2017
> [4] Burda, Edwards, Pathak, Storkey, Darrell, and Efros. Large-scale study of curiosity-driven learning. arXiv preprint
> [5] Nagabandi, Kahn, Fearing and Levine. Neural Network Dynamics for Model-Based Deep Reinforcement Learning with Model-Free Fine-Tuning. ICRA 2018

---

> ### Author Response · Authors · 2018-11-17
> **Response to Reviewer 2**
>
> Thank you for your insightful comments and suggestions! We have made many changes based on the comments provided by reviewers, which are summarized below. We would appreciate it if the reviewer could take another look at our changes and additional results, and let us know if they would like to request additional changes that would alleviate their concerns.
>
> New comparisons: We have added a model-based RL algorithm planning with MPC (Nagabandi et al.), as a comparison to learning features. On the “reach-while-avoid” task (Fig 8), model-based RL struggles compared to a model-free policy with ARC because of challenges such as model-bias, limited exploration and short-horizon planning. The updated plot and corresponding discussion have been added to Section 6.6. We have also added a comparison to representations from inverse dynamics models (Burda et al), described in Section 6.3.
>
> Lack of details: We apologize for the lack of clarity in the submission! We have updated the main text and added an appendix with additional details of the ARC representation and the experimental setup: how a goal-conditioned policy is trained (Sec 6.2, Appendix A.1), how the ARC representation is learned (Sec 6.2, Appendix A.2) , and how the methods are evaluated on downstream applications (Sec 6.5-7, Appendix A.3-6). We have added a discussion of how all comparisons are trained, and measures taken to ensure fairness (Sec 6.3, Appendix A.2, B)
>
> Requirement for goal-conditioned policy: The ARC representation is extracted from a goal-conditioned policy (GCP), requiring us to assume that we can train such a GCP. This assumption was explicit in our submission, but we have emphasized it more now by editing Section 1 and Section 3. For our experiments, the GCP was trained with existing RL methods using a sparse task-agnostic reward (Section 6.2, Appendix A.1) -- obtaining such a policy is not especially difficult, and existing methods are quite capable of doing so [1,2,3].  We therefore believe that this assumption is reasonable, and have added this to the paper in Section 3.
>
> We also ensure that our experiments fairly account for the data required to train the GCP.
> - In the generalization experiment (Section 6.4), all methods initialize behaviour from the GCP, as policies trained from scratch fail, a new comparison we have added to Figure 7.
> - In the hierarchy experiment (Section 6.7), all representations use the GCP as a low-level controller, so ARC incurs no additional sample cost in comparison. Two comparisons (TRPO, Option Critic) which do not use the GCP make zero progress, even when provided with substantially more samples.
> - In the experiment for learning non goal-reaching tasks (Section 6.6), the ARC representation can be re-used across many different tasks without retraining the GCP, amortizing the cost of learning the GCP.  We plan to add an experimental comparison on a family of 100 tasks to demonstrate this amortization, and will update the paper with results.

---

### Official Review · AnonReviewer1 · 2018-11-08
**A good idea, but suffers from lack of clarity**

**Rating:** 5
**Confidence:** 4

**Review:**

The paper suggests a method for generating representations that are linked to goals in reinforcement learning. More precisely, it wishes to learn a representation so that two states are similar if the policies leading to them are similar.

The paper leaves quite a few details unclear. For example, why is this particular metric used to link the feature representation to policy similarity? How is the data collected to obtain the goal-directed policies in the first place? How are the different methods evaluated vis-a-vis data collection?  The current discussion makes me think that the evaluation methodology may be biased. Many unbiased experiment designs are possible. Here are a few:

A. Pre-training with the same data

1. Generate data D from the environment (using an arbitrary policy).
2. Use D to estimate a model/goal-directed policies and consequenttly features F.
3. Use the same data D to estimate features F' using some other method.
4. Use the same online-RL algorithm on the environment and only changing features F, F'.

B. Online training

1. At step t, take action $a_t$, observe $s_{t+1}$, $r_{t+1}$
2. Update model $m$ (or simply store the data points)
3. Use the model to get an estimate of the features

It is probably time consuming to do B at each step t, but I can imagine the authors being able to do it all with stochastic value iteration.

All in all, I am uncertain that the evaluation is fair.

---

> ### Author Response · Authors · 2018-11-17
> **Response to Reviewer 1**
>
> Thank you for your insightful comments and suggestions! We have made many changes based on the comments provided by reviewers, which are summarized below. We would appreciate it if the reviewer could take another look at our changes and additional results, and let us know if they would like to revise their score or request additional changes that would alleviate their concerns.
>
> New comparisons:
>  We have added two more comparisons - with model based RL methods ([1] Nagabandi et al) and learning representations via inverse dynamics models ([2] Burda et al). These have been described in Section 6.3 and added to plots in Fig 7, 8, 10. We have also added a new comparison to learning from scratch for the reward shaping experiment (Section 6.5, Fig 7).
>
> Lack of details:
>  We apologize for the lack of clarity in the submission! We have updated the main text and added an appendix with additional details of the ARC representation and the experimental setup: goal-conditioned policy (GCP) training (Sec 6.2, Appendix A.1), ARC representation learning (Sec 6.2, Appendix A.2) , downstream evaluation (Sec 4, 6.5-6.7, Appendix A.3-6). We have added a discussion of how all comparisons are trained, and measures taken to ensure fairness (Sec 6.3, Appendix A.2, B). We have clarified the algorithm and task descriptions in Section 4 and Section 6.
>
> Fairness of comparisons:
> To ensure the comparisons are fair, every comparison representation learning method is trained using the same data, and we have updated the paper to emphasize this (Section 6.2, 6.3). All representations are trained on a dataset of trajectories collected from the goal-conditioned policy, similar to the (A) scheme proposed by AnonReviewer1. We have updated the paper to include full details of the training scheme for all methods (Section 6.3, Appendix A.2, B).
>
> We also ensure that our experiments fairly account for the data required to train the GCP.
> - In the generalization experiment (Section 6.4), all methods initialize behaviour from the GCP, as policies trained from scratch fail, a new comparison we have added to Figure 7.
> - In the hierarchy experiment (Section 6.7), all representations use the GCP as a low-level controller, so ARC incurs no additional sample cost. Two comparisons (TRPO, Option Critic) which do not use the GCP make zero progress, even with substantially more samples.
> - In the experiment for learning non goal-reaching tasks (Section 6.6), the ARC representation can be re-used across many different tasks without retraining the GCP, amortizing the cost of learning the GCP.  We plan to add an experimental comparison on a family of 100 tasks to demonstrate this amortization, and will update the paper with results.
>
> Find responses to particular questions and comments below:
>
> “How is the data collected to obtain the goal-directed policies in the first place?”
> -> We train a goal-conditioned policy with TRPO using a task-agnostic sparse reward function. We have updated the paper to reflect this (Section 6.2, Appendix A.1).
>
> “why is this particular metric used to link the feature representation to policy similarity?”
> -> We add an explicit discussion of this in Section 3. We link feature representation to policy similarity by this metric, because it directly captures the notion that features should represent elements of the state which directly affect the actions. The KL divergence between policy distributions allows us to embed goal states which induce similar actions similarly into feature space.
>
>
> [1] Nagabandi, Kahn, Fearing and Levine. Neural Network Dynamics for Model-Based Deep Reinforcement Learning with Model-Free Fine-Tuning. ICRA 2018
> [2] Burda, Edwards, Pathak, Storkey, Darrell, and Efros. Large-scale study of curiosity-driven learning. arXiv preprint

---

> > ### Comment · AnonReviewer1 · 2018-11-22
> > **Some clarifications about the experiment design**
> >
> > This looks much better in terms of the details. I think that there's a minor weakness remaining, quite common in many deep learning RL papers: It seems that you are tuning the hyperparameters of the algorithms in the same environments in which they are testing them (you do not specify exactly how the tuning is done). While this is OK for preliminary results, it does have a biasing effect when trying to compare different methods.
> >
> > So, for the moment this appears to be weak evidence in favour of this representation, but it is not entirely convincing.

---

> > > ### Author Response · Authors · 2018-11-24
> > > **Author Response: Adding Clarifications on Hyperparameters**
> > >
> > > Thank you for your response! We are not sure we fully understand your concern about hyperparameter tuning, and were hoping for some additional clarifications regarding this. We have added additional details to the paper regarding hyperparameter tuning in Appendix D. We do not have many hyperparameters to tune for ARCs - the only free parameter is the size of the latent dimension, and for the downstream tasks, we tune the weight of the shaping term for reward shaping and the number of clusters for HRL for each comparison method on each task.
> > >
> > > The size of the latent dimension is selected by performing a sweep on the downstream reward-shaping task for each domain and method. For the reward-shaping task, for each domain and comparison method, the parameter controlling the relative scaling of the shaped reward is selected according to a coarse hyperparameter sweep. The number of clusters for k-means for the hierarchy experiments is similarly selected for each domain and comparison method, although we found that all tasks and methods worked well with the same number of clusters. As you note, this is standard in deep reinforcement learning research, and we are simply following standard practice. Importantly, we give all methods a fair chance by tuning each comparison method separately. While we could certainly adjust hyperparameters differently, we did not find overall that hyperparameters were a major issue for our method. We would appreciate if you could clarify whether you are concerned about this issue in particular, and what a reasonably fair alternative might be?
> > >
> > > If you believe that the issues in the paper have been addressed, we would appreciate it if you would revise your original review, or else point out what remaining issues you see with the paper or experimental evaluation.

---

### Public Comment · (anonymous) · 2018-11-05
**Nice work, though perhaps not very applicable**

This is a nicely written paper, with some interesting and natural ideas about learning policy representations. Simplifying, the main idea is to consider two states $s_1,s_2$ similar if the corresponding policies $\pi_1,\pi_2$ for reaching $s_1, s_2$ are similar.

However, it is unclear how this idea can be really applied when the optimal goal-directed policies are unknown. The algorithm, as given, relies on having access to a simulator for learning those policies in the first place.  This is not necessarily a fatal fault, as long as the experiments compare algorithms in a fair and unbiased manner. How were the data collected in the first place for learning the representations? Was the same data used in all algorithms?

---

> ### Author Response · Authors · 2018-11-20
> **Author Response: Added Clarifications about Fair Comparisons**
>
> Thank you for your interest in our paper and for your insightful comments.
>
> You are correct that the ARC representation requires us to assume that we can train a goal-conditioned policy in the first place. For our experiments, the GCP was trained with TRPO using a sparse reward (see Section 6.2 and Appendix A.1) -- obtaining such a policy is not especially difficult, and existing methods are quite capable of doing so [1,2,3].  We therefore believe that this assumption is reasonable.
>
> To ensure the comparisons are fair, every representation learning method that we compare to is trained using the same data (Section 6.2, 6.3). All representations are trained on a dataset of trajectories collected from the goal-conditioned policy, and we have updated the paper with full details of the training scheme (Section 6.3, Appendix A.2, B).
>
> We also ensure that our experiments fairly account for the data required to train the GCP.
> - In the generalization experiment (Section 6.4), all methods initialize behaviour from the GCP, as policies trained from scratch fail, a new comparison we have added to Figure 7.
> - In the hierarchy experiment (Section 6.7), all representations use the GCP as a low-level controller, so ARC incurs no additional sample cost. Two comparisons (TRPO, Option Critic) which do not use the GCP make zero progress, even with substantially more samples.
> - In the experiment for learning non goal-reaching tasks (Section 6.6), the ARC representation can be re-used across many different tasks without retraining the GCP, amortizing the cost of learning the GCP.  We plan to add an experimental comparison on a family of 100 tasks to demonstrate this amortization, and will update the paper with results.
>
> [1] Nair, Pong, Dalal, Bahl, Lin, and Levine. Visual reinforcement learning with imagined goals.NIPS 2018
> [2] Pong, Gu, Dalal, and Levine. Temporal difference models: Model-free deep rl for model-based control. ICLR 2018
> [3] Andrychowicz, Wolski, Ray, Schneider, Fong, Welinder, P., McGrew, B., Tobin, J., Abbeel, P., and Zaremba, W. (2017). NIPS 2017

---

### Comment · Area_Chair1 · 2018-11-20
**detailed author responses provided; revised version posted; reviewers: please advise further**

Thanks to all for the detailed reviews and review responses.
I could summarize the reviews as: interesting ideas;  needs evaluations that take into account original construction of the goal-directed policies; more details.   The authors have provided detailed responses.
A revised version is available; see the "show revisions" link, for either the revised PDF, or a comparison that highlights the revisions (I can recommend this).

Reviewers (and anonymous commenter), your further thoughts would be most appreciated.
-- area chair

---

### Meta-Review · Area_Chair1 · 2018-12-16
**good idea;  general consensus**

**Confidence:** 4
**Recommendation:** Accept (Poster)

**Metareview:**

To borrow the succinct summary from R1, "the paper suggests a method for generating representations that are linked to goals  in reinforcement learning. More precisely, it wishes to learn a representation so that two states are similar if the
policies leading to them are similar." The reviewers and AC agree that this is a novel and worthy idea.

Concerns about the paper are primarily about the following.
(i) the method already requires good solutions as input, i.e., in the form of goal-conditioned policies, (GCPs)
and the paper claims that these are easy to learn in any case.
As R3 notes, this then begs the question as to why the actionable representations are needed.
(ii) reviewers had questions regarding the evaluations, i.e., fairness of baselines, additional comparisons, and
additional detail.

After much discussion, there is now a fair degree of consensus.  While R1 (the low score) still has a remaining issue with evaluation, particularly hyperparameter evaluation, they are also ok with acceptance. The AC is of the opinion that hyperparameter tuning is of course an important issue, but does not see it as the key issue for this particular paper.
The AC is of the opinion that the key issue is issue (i), raised by R3. In the discussion, the authors reconcile the inherent contradiction in (i) based on the need of additional downstream tasks that can then benefit from the actionable representation, and as demonstrated in a number of the evaluation examples (at least in the revised version). The AC believes in this logic, but believes that this should be stated more clearly in the final paper. And it should be explained
the extent to which training for auxiliary tasks implicitly solve this problem in any case.

The AC also suggests nominating R3 for a best-reviewer award.